# Sulfatase 2 Affects Polarization of M2 Macrophages through the IL-8/JAK2/STAT3 Pathway in Bladder Cancer

**DOI:** 10.3390/cancers15010131

**Published:** 2022-12-26

**Authors:** Wentao Zhang, Fuhan Yang, Zongtai Zheng, Cheng Li, Shiyu Mao, Yuan Wu, Ruiliang Wang, Junfeng Zhang, Yue Zhang, Hong Wang, Wei Li, Jianhua Huang, Xudong Yao

**Affiliations:** 1Department of Urology, Shanghai Tenth People’s Hospital, School of Medicine, Tongji University, Shanghai 200070, China; 2Urologic Cancer Institute, School of Medicine, Tongji University, Shanghai 200070, China; 3Department of Urology, Hefei Cancer Hospital, Chinese Academy of Sciences, Hefei 230031, China; 4Department of Central Laboratory, Clinical Medicine Scientific and Technical Innovation Park, Shanghai Tenth People’s Hospital, Shanghai 200435, China

**Keywords:** SULF2, bladder cancer, macrophages

## Abstract

**Simple Summary:**

At present, the immunotherapy and immune microenvironment of bladder cancer have attracted more and more attention from clinicians and researchers. In this study, the high expression of SULF2 by RNA sequencing of tissues from 90 bladder cancer patients was associated with poor prognosis in patients. We further found that a high expression of SULF2 can promote the polarization of macrophages to M2. In mechanistic studies, we found that SULF2 can promote the secretion of IL-8 through Wnt/β-catenin. IL-8 promotes the transcription of CD163 and CD206 in the microenvironment, ultimately leading to the polarization of macrophages to M2 macrophages. This study elucidates the effect of SULF2 on the polarization of macrophages in the tumor microenvironment.

**Abstract:**

Sulfatase 2 (SULF2) affects the occurrence and development of cancer by regulating HSPG-binding factors. However, the mechanism of SULF2 in bladder cancer (BCa) is unknown. To determine this, we analyzed the RNA sequencing of 90 patients with BCa. The results showed that the expression of SULF2 was closely related to the prognosis of BCa. Moreover, in vivo and in vitro experiments revealed that SULF2 promotes tumor proliferation and invasion. Furthermore, using a mouse orthotopic BCa model and flow cytometric analysis, we identified that SULF2 affects the polarization of macrophages. Mechanism studies clarified that SULF2 promoted the release of HSPG-binding factors, such as IL-8, in the microenvironment through β-catenin. Meanwhile, IL-8 activated the JAK2/STAT3 pathway of macrophages to promote the expression of CD163 and CD206, thereby regulating the polarization of macrophages to the M2-type. Conclusively, these results indicate that SULF2 plays an important role in regulating the microenvironment of BCa and promotes the polarization of macrophages to the M2-type by secreting IL-8, which further deepens the malignant progression of BCa.

## 1. Introduction

Bladder cancer (BCa) is the most common malignant tumors of the urinary system and ranks four among the most common malignant tumors in men worldwide. It is estimated that nearly 40,000 patients die from BCa every year, and 549,393 new cases were reported in 2018 [1]. Bladder cancer can be divided into nonmuscle invasive bladder cancer (NMIBC) and muscle-invasive bladder cancer (MIBC) according to the depth of tumor invasion. In the past 30 years, there have been few advances in the research and treatment of BCa. With the understanding of the biological functions and immune microenvironment of BCa coupled with large-scale gene sequencing, effective targeted therapies and immunotherapies are used for BCa [2,3].

Currently, the role of the tumor microenvironment (TME) in the progression and metastasis of BCa has received tremendous attention [4]. The TME includes tumor cells, macrophages, fibroblasts, endothelial cells, and noncellular substances [5]. The response of macrophages to microenvironmental signals is usually polarized into two phenotypes, namely M1 and M2 macrophages. M1 macrophages have antitumor and immune promotion effects. In contrast, M2 macrophages promote tumor immune escape, angiogenesis, tumor growth, and metastasis [6,7]. Hence, it is important to explore the molecular mechanism of the communication between tumor cells and macrophages in the microenvironment of BCa.

Heparin sulfate proteoglycan (HSPG) binds to various ligands, such as heparin-binding growth factor, fibroblast growth factor, and vascular endothelial growth factor, to perform a wide variety of biological functions [8,9]. Recent studies showed that the proteoglycan structure undergoes profound aberrations, and this leads to uncontrolled tumor proliferation, immune escape, metastasis, and differentiation of tumors, such as rectal cancer [10]. These structural modifications are mainly attributed to abnormal changes in the expression of heparin sulfate biosynthetic or catabolic enzymes, such as sulfatase. SULF1 and SULF2 belong to the Sulfatases family. SULF1 and SULF2 are extracellular heparan sulfate 6-O-endosulfatases with unique structural characteristics, enzyme activities, and signaling functions [11]. Studies have shown that SULF1 can bind to FGF, HGF, and VEGF and regulate the proliferation of a variety of solid tumors, including bladder cancer [12,13]. SULF2 also can modulate the expression of cell surface signal molecules on tumor cells by regulating the sulfation of the glycosyl part of HSPGs. Presently, studies have indicated that SULF2 is a potential tumor driver gene for cholangiocarcinoma and glioblastoma [14,15]. However, its role in BCa is yet unclear. Therefore, in this study, we aimed to investigate the mechanism by which overexpression of SULF2 in BCa cells promotes TME BCa progression.

## 2. Materials and Methods

### 2.1. Patient Collection

Ninety BCa tissues were collected from patients who underwent either TURBT or radical cystectomy at the Shanghai Tenth People’s Hospital (STPH) from November 2019 to September 2020. Prior informed consent was obtained from the patients (SHSY-IEC-4.1/19-120/01).

The protocols of total RNA extraction, paired-end libraries synthetization, and RNA sequence were available in our previous methods [16]. The mRNA (RNA sequencing) fragments per kilobase of transcripts per million mapped reads (FPKM) standardized values of genes were calculated and selected for further analyses. Monthly telephone follow-up was done to obtain the disease-free survival (DFS) of these patients.

The FPKM standardized mRNA expression data and clinical information of BLCA patients were downloaded from the Cancer Genome Atlas (TCGA, https://gdc-portal.nci.nih.gov/) (Accessed on 9 January 2020). The processed gene expression data of GSE13507, GSE48277, GSE32894, and GSE31684 with the prognostic information of BLCA patients were downloaded from the Gene Expression Omnibus (GEO, https://www.ncbi.nlm.nih.gov/) (Accessed on 9 January 2020). The processed gene expression data and clinical information of IMvigor210 trial that included patients with metastatic urothelial cancer treated with atezolizumab (PD-L1 inhibitor) were obtained from the website http://research-pub.gene.com/IMvigor210CoreBiologies (Accessed on 9 January 2020). The clinical characteristics of BCa patients in each cohort are shown in Appendix A.

### 2.2. Cell Lines

Human BCa cell lines, namely T24, UMUC3, and J82, normal urothelial cell line SV-HUC-1, mouse BCa cell line MB49, and human acute monocyte THP-1 leukemia cells were purchased from Chinese Academy of Sciences, Institute of Chemistry and Cell Biology (Shanghai, China). Cell lines were cultured in either DMEM or RPMI-1640 medium with 10% fetal bovine serum (FBS) (Gibco, Waltham, MA, USA) and 1% penicillin/streptomycin (Hyclone; GE Healthcare Life Sciences, Logan, UT, USA). All cells were cultured in a 37 °C incubator with 5% CO_2_.

### 2.3. Cell Proliferation Assay 

The effect of SULF2 on cell proliferation was assessed by CCK-8 assay and 5-ethynyl-20-deoxyuridine (EDU) incorporation experiment. For the EDU assay, the experiment was carried out according to the manufacturer’s instructions. For the CCK-8 assay, cells were seeded into 96-well plates at a density of 2 × 10^3^ cells/well, 10 μL of CCK-8 reagent (Yeasen Biotechnology, Shanghai, China) was added in each well, and the cells were maintained at 37 °C for 1.5 h. The absorbance was measured at 450 nm using a microplate spectrophotometer. Each experiment was performed in triplicate.

### 2.4. Western Blot and Co-IP

Using RIPA lysis buffer, we extracted total protein from cells or tissues and then quantified the lysate by the bicinchoninic acid (BCA) protein assay. Next, 30 μg protein sample was subjected to sodium dodecyl sulfate-polyacrylamide gel electrophoresis (SDS-PAGE), and then proteins were transferred to nitrocellulose membrane (Sigma-Aldrich; Merck KGaA). The membrane was blocked with 5% skim milk at room temperature for 1 h, and then immunoblotting was performed with primary antibody at 4 °C overnight. Subsequently, the membrane was incubated with the HRP-labeled secondary antibody for 1 h. The Super ECL Detection (Yeasen Biotech Co. Ltd., Shanghai, China) was used to visualize and quantify the immunoreactive bands. The results were visualized and quantified by Tanon 5200 system (Tanon, Shanghai, China). GAPDH was used as an internal reference. Antibodies are listed in Appendix A. For Co-IP, lysates of 1 × 10^7^ Bca cells were immunoprecipitated with IP buffer containing IP antibody-coupled argarose beads, and the complexes were used to western blot. IgG was used as a negative control. Original blots see Appendix A.

### 2.5. RNA Extraction and Real-Time Polymerase Chain Reaction (qPCR)

TRIzol reagent (Invitrogen, Carlsbad, CA, USA) was used to extract total RNA from human tissues or cells according to the manufacturer’s instructions. A reverse transcription system kit (Vazyme Biotech Co., Ltd., Shanghai, China) was used to generate the first strand of cDNA. The ChamQ Universal SYBR qPCR Master Mix and ABI Prism 7500 sequence detection system (Applied Biosystems, Foster City, CA, USA) were used for qPCR. The qPCR parameters were as follows: 40 cycles of 30 s at 95 °C, then 10 s at 95 °C, and 30 s at 60 °C. GAPDH was used as an endogenous control (Appendix A). The relative fold change was analyzed by the 2-ΔΔC(t) method. Each experiment was performed in triplicate.

### 2.6. Reagents and Enzyme-Linked Immunosorbent Assay (ELISA) Assay

Human recombinant interleukin 8 (IL-8; PeproTech, Rocky Hill, NJ) was dissolved in trehalose at a concentration of 100 μg/mL and stored at −20 °C. At the time of use, the final concentration of IL-6 was adjusted to 100 ng/mL in the appropriate medium. AG490 (MedChemExpress, Monmouth Junction, NJ, USA), a tyrosine kinase inhibitor that can inhibit the signal transducer and activator of transcription 3 (STAT3) signaling pathway, was dissolved in dimethyl sulfoxide at a concentration of 5 mg/mL, stored at −20 °C, and protected from light. The THP-1 cells were seeded at a density of 3 × 10^6^ cells/flask and exposed to 100 ng/mL phorbol 12-myristate 13-acetate (PMA; Sigma-Aldrich, Burlington, MA, USA) for 48 h to obtain macrophage-like differentiated THP-1 cells. Then, the medium containing PMA was replaced with fresh medium to obtain resting state of macrophages (M0). Next, to differentiate into M1 phenotype, we added 20 ng/mL IFNγ and 1 mg/mL LPS (Sigma-Aldrich), and for M2 phenotype we added 20 ng/mL interleukin 4 (IL-4; PeproTech, Rocky Hill, NJ, USA).

The Human Cytokine Elisa kit (Zorin Biological, Shanghai, China) was used to detect the concentration of the supernatant cytokines in the coculture system. Cytokine and chemokine levels were determined with standard curves developed for each experiment according to the standards provided by the manufacturer. The relative expression levels were calculated from the experimental results.

### 2.7. Animal Models

For orthotopic bladder cancer model: C57 female mice were used for in situ injection of MB49 BCa cells, and 1 × 10^6^ treated cells were injected into the bladder through the urethra, five in each group (NC and shSULF2). After four weeks, the mice were sacrificed, and in situ tumors were taken out and immunized by flow cytometry cell infiltration situation. 

For subcutaneous xenotransplant tumors: 1 × 10^6^ treated T24/UMUC3 cells were injected into the right armpit of four-week-old female BALB/c nude mice, five in each group. After four weeks, the mice were sacrificed, and the subcutaneous tumors were removed. The mouse body weight, tumor diameter, and tumor volume were monitored every seven days as per the formula (tumor volume = π/6 × length × width^2^). 

For tail vein lung metastasis model: 1 × 10^6^ cells with either overexpression or knockdown of SULF2 were injected into the tail vein of four-week-old nude mice, with six mice in each group. The in vivo imaging system (IVIS) was used weekly after three weeks to observe the tumor metastasis.

### 2.8. Flow Cytometry

To analyze the immune cells infiltrating the mouse orthotopic BCa model, the mice were sacrificed after four weeks of MB49 injection into the C57 mice bladder. The tumor tissues were taken out, cut, and incubated with collagenase (100 μg/mL) at 37 °C for 30 min. The separated tumor cells were filtered through a 70 µm mesh, and red blood cells were removed using red blood cell lysis buffer. The tumor cells were resuspended in PBS and then incubated with flow cytometry antibodies. After 30 min, the cells were washed thrice with PBS followed by resuspension in 500 µL of PBS buffer for flow cytometry analysis. For gating strategy, live–dead dye (APC-Cy7) can separate living cells. Then, CD45 (FITC) can separate immune cells from living cells, CD11b (percp5.5) and F4/80 (BV421) can separate macrophages, CD86 (APC) can separate M1 macrophages, and CD206 (PE) can separate M2 macrophages.

For cell cycle analysis, the tumor cells were collected, trypsinized to form a cell suspension, and then fixed with 75% ethanol at 4 °C overnight. Next day, the cells were washed thrice with PBS followed by incubation with RNase containing iodide (PI, 40%, Sigma-Aldrich) for 30 min and then washed thrice with PBS and used for flow cytometry (BD Biosciences, Franklin Lakes, NJ, USA). The experiment was repeated thrice. The results were analyzed using FlowJo. 

### 2.9. Establishment of Stable Cell Lines

The specific shRNAs targeting SULF2 were cloned into the lentivirus vector pLKO (Zorin, Shanghai, China) (Appendix A). The full-length cDNA amplified by PCR was inserted into the lentivirus vector pCDH (Zorin, Shanghai, China) to produce a specific plasmid overexpressing SULF2. Then, 293T cells were transfected with specific sequence pCDH or pLKO and packaging plasmid. Virus particles were generated and collected 48 h after transfection. The virus was used to infect bladder cancer cell lines for 24 h and then used puromycin to screen stable cell lines.

### 2.10. Cocultivation System

The BCa cells and macrophages were cocultured using a 0.4 µm Transwell chamber (Corning, Lowell, MA, USA). The THP1 cell line was placed in the lower chamber while BCa cell line with either overexpression or knockdown of SULF2 was placed in the upper chamber. The cocultivation time was 48 h. The required cells were collected for the next experiment.

### 2.11. Colony Formation

For colony formation analysis, cell lines with either SULF2 overexpression or knockdown were seeded into a six-well plate at a density of 1 × 10^3^ cells/well and cultured in an incubator for 14 days. After this, the cells were washed thrice with cold PBS, fixed with 75% ethanol, and stained with 0.1% crystal violet. The images of the stained tumor cell colony were recorded with a digital camera and statistically analyzed.

### 2.12. Cell Migration and Invasion

The cell migration ability was detected using a Transwell chamber (Corning, Lowell, MA). Then, 5 × 10^4^ cells were added to 200 µL of FBS-free RPMI-1640 medium and transferred to the upper chamber of the transwell while 600 μL of cell culture medium containing 10% FBS was added to the lower chamber. The transwell chamber was incubated at 37 °C for 16 h and then washed thrice with cold PBS. Cells that did not pass through the upper chamber were carefully wiped with a wet cotton swab, fixed with 75% ethanol for 30 min, and then the chamber stained with 0.1% crystal violet for 10 min. The cell migration was observed under a microscope (Olympus Corporation). For tumor cell invasion experiments, the Transwell chambers were precoated with Matrigel, and the remaining experimental procedure was same as the tumor cell migration experiment. Each experiment was performed in triplicate.

### 2.13. Immunohistochemistry (IHC)

The human BCa tissues and mouse bladder cancer tissues were fixed in cold 4% paraformaldehyde, embedded in paraffin, and sectioned. After hydration, antigen retrieval, and blocking procedures, the tissues were incubated with the antibody overnight at 4 °C. Next, the sections were incubated with biotinylated goat antirabbit antibody IgG for 20 min at room temperature and then incubated with streptavidin-horseradish peroxidase for 30 min. Subsequently, diaminobenzidine-H_2_O_2_ was used as a substrate for peroxidase. When the cytoplasm of the cancer cells was stained yellow, two pathologists observed the pathological sections and confirmed a positive result.

### 2.14. Statistical Analysis

SPSS version 22 (SPSS, Inc., Chicago, IL, USA) and R statistical software (https://www.r-project.org/) (Accessed on 9 January 2020) were used for data analysis. The Kaplan-Meier (log-rank test) was used to compare the overall survival (OS) and DFS of the patients with BCa between low and high expression groups of SULF2. Patients were divided into low and high SULF2 expression groups based on the optimal cutoff values in each dataset calculated by the ‘survminer’ R package v0.4.8. The differences between the groups were evaluated using Student’s *t*-test, chi-square test, and the Mann–Whitney U test, as appropriate. One-way ANOVA followed by Bonferroni test was used to compare multiple groups. A *p*-value of <0.05 was considered statistically significant.

## 3. Results

### 3.1. SULF2 was Highly Expressed in BCa and Associated with Poor Prognosis of Patients

We extracted RNA from the tumor and the adjacent tissues of 20 patients with BCa. Using qPCR, we found that the expression of SULF2 was significantly high in cancer tissues (Figure 1A). The RNA sequencing results of cancer tissues of 90 patients showed that patients with MIBC had higher SULF2 expression than those with NMIBC (*p* = 0.003) (Figure 1B). Moreover, patients with BCa in the high SULF2 expression group had poorer DFS than those in the low SULF2 expression group (*p* = 0.038) (Figure 1C). In addition, Kaplan–Meier analysis and univariate Cox analyses in the TCGA, IMvigor210 trial, and GSE32894 showed that SULF2 expression was an adverse prognostic factor of patients with BCa (Figure 1D–F).

Further, a meta-analysis of six studies involving 1308 patients with BCa revealed that SULF2 expression was significantly associated with poor OS (random effects model, HR = 1.64, 95% CI = 1.14–2.36). No publication bias was measured by Egger’s funnel plot (*p* > 0.05) (Figure 1G,H).

Further, we verified the expression of SULF2 in BCa tissues and adjacent tissues through IHC and western blot and found that SULF2 was highly expressed in BCa tissues (Figure 2A,B). Additionally, we verified the expression of SULF2 in BCa cell lines by qPCR and western blot and confirmed that SULF2 expression was higher in a variety of BCa cell lines when compared with the immortalized normal urothelial cells (SV-HUC-1). Thus, we selected T24 and UMUC3 for further experiments (Figure 2C,D). 

### 3.2. SULF2 Affects the Proliferation, Migration, and Invasion of BCa Cells

We evaluated the effect of SULF2 expression on the proliferation of BCa cells by either overexpression or knockdown of SULF2. We knocked down the expression of SULF2 in the T24 cell line and selected shRNA-#2 since it had the highest knockdown efficiency among the three shRNAs to construct stable transgenic strains through lentivirus. At the same time, we constructed a stable transgenic strain overexpressing SULF2 in the UMUC3 cell line through lentivirus (Figure 2E–G). 

For in vivo experiments, we subcutaneously injected T24 or UMUC3 stable transgenic cells into nude mice to establish a xenograft mouse model. We observed that tumors in the T24 group of sh-SULF2 were significantly small while tumors in the UMUC3 group of oe-SULF2 were significantly large (Figure 3A–C and Appendix A). Additionally, IHC results of subcutaneous tumor tissues showed that in contrast to the oe-SULF2 group, the expression of SULF2 and proliferation marker Ki67 was significantly reduced in the sh-SULF2 group (Figure 3E,F and Appendix A). Moreover, CCK-8, EDU, and colony formation assays found that knockdown of SULF2 significantly inhibited cell proliferation whereas overexpression significantly promoted cell proliferation (Figure 3D,G,H).

The effect of SULF2 on the migration and invasion of BCa cells was assessed by transwell assay. The results showed that migration and invasion were significantly low in the sh-SULF2 T24 cells (Figure 4A) while significantly high in the oe-SULF2 UMUC3 cells when compared with the negative control (Figure 4B,C). Meanwhile, EMT-related markers were detected by western blot, and the results showed that E-cadherin (epithelial cell marker) was highly expressed in the sh-SULF2 while N-cadherin and vimentin (mesenchymal cell marker) decreased. (Figure 4D). Flow cytometry analysis showed that the knockdown of SULF2 caused S-phase arrest (Figure 4E,F). In addition, we established a tail vein metastasis model. IVIS showed that knockdown of SULF2 expression significantly inhibited the size and number of tumor metastases while overexpression of SULF2 promoted tumor metastasis (Figure 4G and Appendix A). Figure 4H shows the relative fluorescence intensity of the tail vein model.

### 3.3. SULF2 Promotes Differentiation of Macrophages into M2-Type in BCa Microenvironment

Previous studies indicated that SULF2 affects the microenvironment [17,18]. Therefore, in the present study, we investigated whether SULF2 affects the BCa microenvironment. Bioinformatics analysis showed that macrophage expression is significantly abnormal in the BCa microenvironment. According to the TCGA database, M2 macrophages are significantly higher in BCa tumors than in normal tissues (Figure 5A,B). GSEA analysis of the STPH database showed that several macrophage-related signaling pathways were significantly enriched in the high SULF2 expression group (Figure 5C). Furthermore, upon analyzing the relationship between SULF2 expression in BCa and immune cells in the TME of BCa, we found that SULF2 expression is positively related to M2 macrophages and negatively with M1 macrophages in STPH and TCGA database (Figure 5D,E and Appendix A). In addition. As shown in Figure 5F, we constructed a SULF2 knockdown stable transgenic cell line of MB49 mouse cells and constructed a bladder orthotopic cancer model in C57 mice (Appendix A). Flow cytometry demonstrated that the proportion of M2 macrophages was significantly downregulated after the knockdown of SULF2 (Figure 5G). The percentage of M2 macrophage decreased, and the M1/M2 radio increased in the sh-SULF2 group (Figure 5H,I). Meanwhile, we took the orthotopic cancer tissue for IHC analysis, which indicated a decrease in M1 (CD86) and M2 macrophage markers (CD163 and CD206) (Figure 5J). Therefore, we constructed a coculture system (Figure 6A) where we cocultured knockdown or overexpression cell lines with THP1 and found changes in CD163 and CD206 (Figure 6B). In addition, qPCR also confirmed that the expression of ARG1 (M2 macrophage) significantly reduced while the expression of NOS2 (M1 macrophage) increased (Appendix A). In all, these results suggest that SULF2 may regulate the changes in the BCa microenvironment.

### 3.4. SULF2 Affects the Secretion of IL-8 through β-Catenin

To find the mechanism by which SULF2 affects the polarization of macrophages, we analyzed the expression of IL-6, IL-8, IL10, CXCL1, and other cytokines by ELISA after knockdown and overexpression of SULF2. The results revealed that IL-8 was significantly downregulated in the sh-SULF2 cells while significantly upregulated in the oe-SULF2 cells (Figure 6C,D). qPCR of mouse orthotopic bladder cancer tissue confirmed that the expression of IL8 decreased significantly after the knockdown of SULF2 (Appendix A). The TCGA database also confirmed that SULF2 and IL-8 were significantly positively correlated (Figure 6G and Appendix A). Next, we added IL-8 to the sh-SULF2 and oe-SULF2 cells and cocultured them with THP1. Interestingly, we found that the exogenous addition of IL-8 promoted the expression of CD163 and CD206, whether for the knockdown group or overexpression group (Figure 6E,F).

Through GSEA analysis of the RNA sequence we found that SULF2 and Wnt/β-catenin pathways are closely related (Figure 6G and Appendix A). According to previous studies, SULF2 binds to Wnt3a to prevent β-catenin degradation and recruits other transcription coactivators to promote transcription and translation of downstream genes [19]. Therefore, we hypothesized that β-catenin regulated by SULF2 could promote the secretion of IL-8. We found that β-catenin decreased significantly after SULF2 knockdown. However, β-catenin was upregulated in SULF2-overexpressing UMUC3 cell lines (Appendix A). The interaction between SULF2 and Wnt3a in BCa was verified by co-IP (Appendix A). Then, we knocked down β-catenin expression by siRNA in T24 cells and cocultivated them with M0 cells induced by THP1 cells. We found that the knockdown of β-catenin downregulated the expression of CD163 and CD206 (Figure 6H).

### 3.5. SULF2 Promotes Macrophage Polarization through the IL-8/JAK2/STAT3 Pathway

Previous studies reported that the STAT family is critical for the polarization of macrophages [20,21]. Moreover, GSEA suggests that SULF2 may affect the JAK-STAT pathway (Figure 7A). Therefore, to study the mechanism by which SULF2 affects the polarization of macrophages, we cocultured cells with knockdown or overexpression of SULF2 with THP1 cells and evaluated the phosphorylation levels of STAT1, STAT5, STAT3, and STAT6 (Appendix A). We found that STAT3 phosphorylation decreased significantly after SULF2 knockdown. Therefore, we suggested that SULF2 affects STAT3 expression in macrophages by regulating the secretion of IL-8. At the same time, we verified that JAK2 phosphorylation was significantly reduced in the coculture system with SULF2 knockdown (Figure 7B).

To further verify, we added IL-8 exogenously to THP1 cells. The results showed that the phosphorylation levels of JAK2 and STAT3 increased significantly (Figure 7C,D). Upon adding STAT3 inhibitor (AG490), we found that C163 and CD206 were significantly inhibited (Figure 7E,F). Collectively, these data indicate that SULF2 reveals a new way for tumor immune escape during BCa progression (Figure 7G).

## 4. Discussion

Macrophages are the main components of the TME. They are transformed into different states due to differences in the microenvironments, such as the increase or decrease of local cytokines, hypoxia, necrosis, or changes in metabolites [22]. Currently, the polarization of macrophages can be divided into the M1-type (classically activated macrophages) and the M2-type (alternatively activated macrophages), which have differential effects on tumor progression. M2 macrophages are closely related to the malignant progression of tumors and are associated with reduced survival rates in pancreatic cancer, kidney cancer, and breast cancer [23,24,25].

In this study, we found that there were significant differences in M2 macrophages in the immune cell infiltration of the BCa microenvironment. This was also confirmed by tumor formation in situ in mice. Additionally, we found that the knockdown of SULF2 in the MB49 cells decreased the proportion of M2 macrophages in the BCa microenvironment. Xu et al.’s research on the analysis of immune cells in BCa also corroborated the phenomenon of increased macrophages in BCa [26]. Our previous clinical study on SULF2 showed that IHC analysis and clinical prognostic follow-up of SULF2 expression in 203 patients with bladder cancer confirmed that SULF2 is highly expressed in bladder cancer and may be associated with lymphatic metastasis, which has clinical significance for evaluating prognosis. In addition, several studies have shown that adiponectin, an inhibitor of SUFL2, may be a potential tumor therapeutic drug [27]. The role and mechanism of adiponectin in bladder cancer still need further verification.

SULF2 not only changes the function of HSPG by regulating 6-O-sulfation but also plays an important role in the occurrence and development of tumors, such as participating in the tumor cell cycle, proliferation, apoptosis, and other biological functions [28]. Therefore, we performed RNA sequencing of BCa tissues obtained from 90 patients with BCa in our hospital. Upon analysis, we found that SULF2 expression was significantly increased, and it is closely related to the OS and PFS of patients with BCa. However, the role and function of SULF2 in the TME of BCa have not been studied yet. Biroccio et al. found that the TSS upstream region of SULF2 contains a binding site for TRF2. The combination of TRF2 and its target motif enhances the promoter activity, and ultimately increases the expression of SULF2, which further promotes VEGF-A secretion and thus affects angiogenesis in the TME [29]. Another study reported that M2 macrophages increase the transcription levels of SULF1 and HSPG2 in bone marrow fibroblasts and affect the microenvironment of prostate cancer bone metastasis [30].

As per previous studies, SULF2 affects the Wnt pathway [31,32]. We also confirmed this in our experiments. Meanwhile, using ELISA, we detected changes in cytokines, including those considered HSPG-binding factors, such as IL6, IL8, and IL10. We found that IL-8 secretion was reduced after SULF2 knockdown. Therefore, we speculated that SULF2 may affect the secretion of IL-8 through β-catenin. IL-8, also known as a leukocyte chemical attractant, recruits macrophages into the TME in a paracrine manner [33,34]. Interestingly, we found that tumor-derived IL-8 promotes phenotypic conversion during the recruitment process to promote M2 macrophage polarization. Upon adding IL-8 to THP1 cells, the expression of CD206 and CD163 increased. These results are consistent with previous studies on IL-8 in liver cancer, oral squamous cell carcinoma, and papillary thyroid cancer [35,36]. Therefore, SULF2 is highly expressed in BCa and increases IL-8 secretion, which promotes the polarization of macrophages to the M2-type and enhances the malignant progression of the tumor through the communication between BCa cells and macrophages.

The JAK/STAT signal transduction pathway is stimulated by cytokines. It is involved in important biological processes, such as proliferation, differentiation, apoptosis, and immune regulation. Ying et al. found that broussonin E can promote the phenotypic transition of macrophages by inhibiting ERK and p38 MAPK and activating the JAK2/STAT3 pathway [37]. In mastitis, researchers identified a crosstalk signaling pathway between breast epithelial cells and macrophages. The exosomal miR-221 promotes the polarization of macrophages through SOCS1/STATs [38]. In this study, we tested the activities of STAT1, STAT3, STAT5a, and STAT6 after cocultivation in THP1 cells. We found that STAT3 was activated in THP1, and the secretion of IL-8 regulated by SULF2 promotes the phosphorylation of JAK2 (Tyr1008) and STAT3 (Tyr705). These results indicate that the effect of IL-8 on promoting the polarization of M2 macrophages depends on the regulation of the JAK2/STAT3 signaling pathway.

## 5. Conclusions

In summary, we identified a novel mechanism of crosstalk between BCa and macrophages in the TME through in vivo and in vitro experiments and bioinformatics verification. SULF2 is highly expressed in BCa and promotes IL-8 (an HSPG-binding factor) secretion through β-catenin. IL-8 secreted into the microenvironment activates the JAK2/STAT3 pathway and leads to the upregulation of CD163 and CD206 transcription that convert macrophages to the M2 phenotype and thus promote malignancy of BCa progression. In the future, the screening and validation of SULF2 inhibitors may require further investigation.

## Figures and Tables

**Figure 1 cancers-15-00131-f001:**
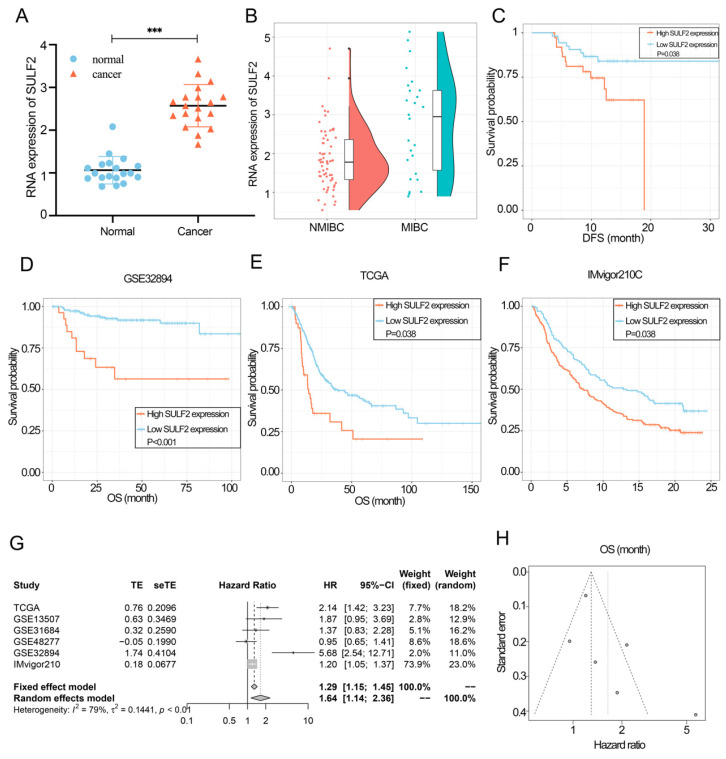
SULF2 is highly expressed in bladder cancer (BCa) and is associated with a poor prognosis. (**A**) SULF2 mRNA was detected in 20 pairs of BCa and adjacent normal tissues using qPCR. (**B**) The difference in SULF2 expression of NMIBC and MIBC was observed by RNA sequencing of BCa tissues obtained from 90 patients with BCa. (**C**) Kaplan–Meier analysis of DFS for 90 patients with BCa with low or high SULF2 expression level. (**D**–**F**) Kaplan–Meier analysis of OS for patients BCa with low or high SULF2 expression level in GSE32894, TCGA, and IMvigor210C dataset. (**G**,**H**) Meta-analysis verified the correlation between SULF2 expression and OS in six datasets. Data are represented as mean ± SD of three experiments. *** *p* < 0.001.

**Figure 2 cancers-15-00131-f002:**
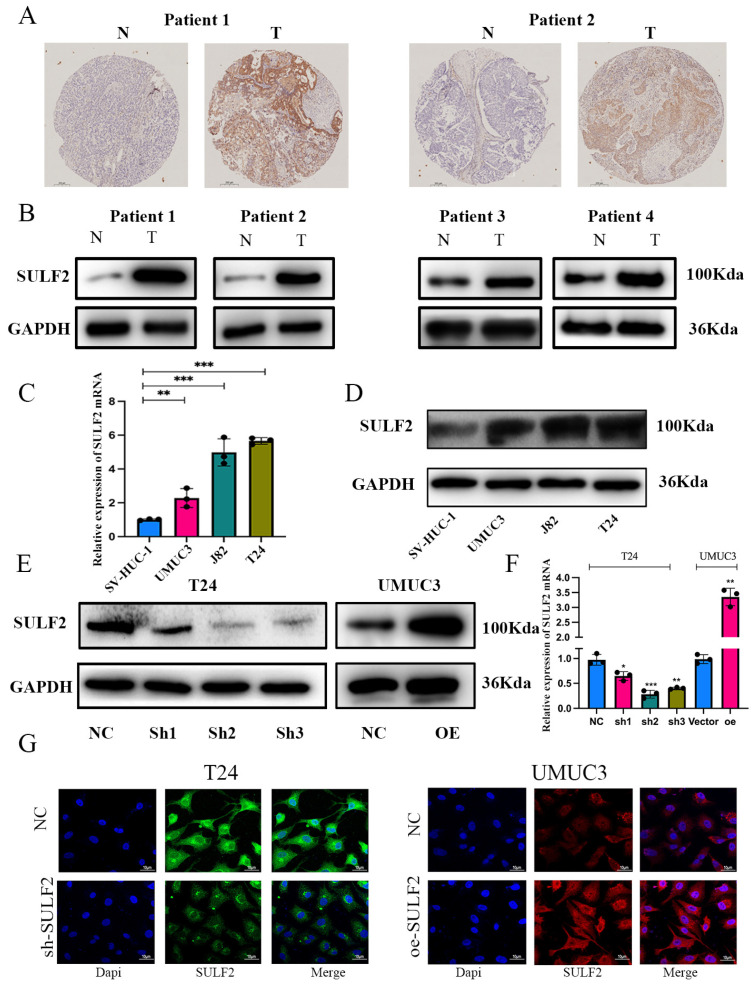
SULF2 is overexpressed in human BCa tissues and cell lines. (**A**) Representative images of SULF2 in BCa tissues (T) and adjacent normal tissues (N) by IHC staining. Scale bar = 200 μm. (**B**) SULF2 expression in BCa tissues and adjacent normal tissues detected by western blot. (**C**) Relative mRNA and (**D**) protein levels of SULF2 in BCa cell lines and the immortalized human normal bladder epithelial cell line SV-HUC-1 detected by qPCR and western blot, respectively. (**E**,**F**) Knockdown and overexpression efficiencies of SULF2 were examined by qPCR and western blot, and GAPDH served as the internal control. (**G**) Immunofluorescence to verify the efficiency of knockdown and overexpression of SULF2. Data are represented as mean ± SD of three experiments. * *p* < 0.05, ** *p* < 0.01, *** *p* < 0.001.

**Figure 3 cancers-15-00131-f003:**
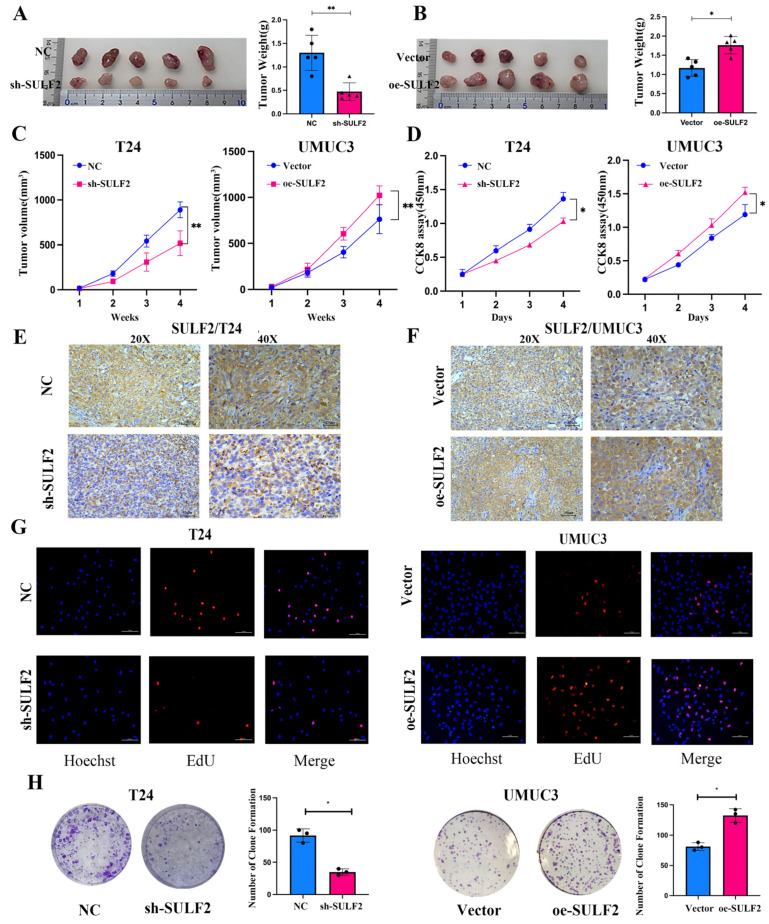
SULF2 affects proliferation of BCa cells. (**A**,**B**) For nude mouse xenograft assay, sh-SULF2, oe-SULF2, and negative control BCa cells were subcutaneously injected into nude mice and monitored for one month for tumor cell xenograft formation and growth. The right is tumor cell xenograft weight. (**C**) Analysis of tumor growth for the subcutaneous xenotransplant tumor models. (**D**) Cell proliferation was detected by CCK8 assay at 24, 48, 72, and 96 h. (**E**,**F**) Representative images of SULF2 in tumor cell xenograft by IHC staining. (**G**) Representative images of EDU assay of sh-SULF2, oe-SULF2, and negative control BCa cells. (**H**) Knockdown or overexpression of SULF2 affects the cell colony formation ability. The right panel shows the number of colonies of cloned cells. Data are represented as mean ± SD of three experiments. * *p* < 0.05, ** *p* < 0.01.

**Figure 4 cancers-15-00131-f004:**
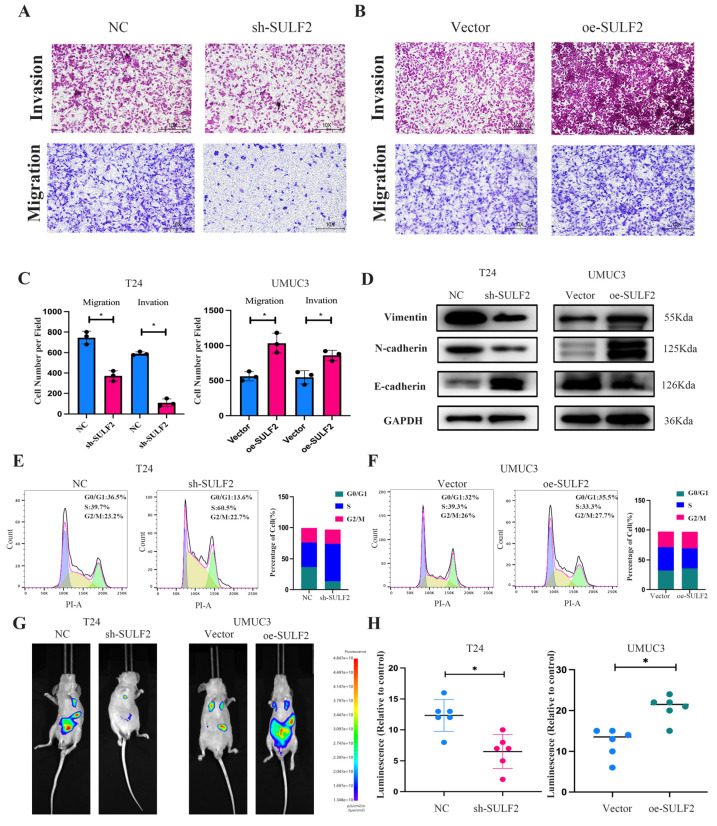
SULF2 affects metastasis of BCa cells. (**A**–**C**) Transwell assay was used to examine the migratory and invasive capacities of BCa cells after knockdown and overexpression of SULF2. Scale bar = 100/50 μm. (**D**) The expression of EMT-associated markers was analyzed by western blot after knockdown and overexpression of SULF2. (**E**,**F**) Flow cytometric detection of the effect of SULF2 on the cell cycle. (**G**,**H**) Representative images of IVIS after tail vein injection of knockdown or overexpressing SULF2 cells. Data are represented as mean ± SD of three experiments. * *p* < 0.05.

**Figure 5 cancers-15-00131-f005:**
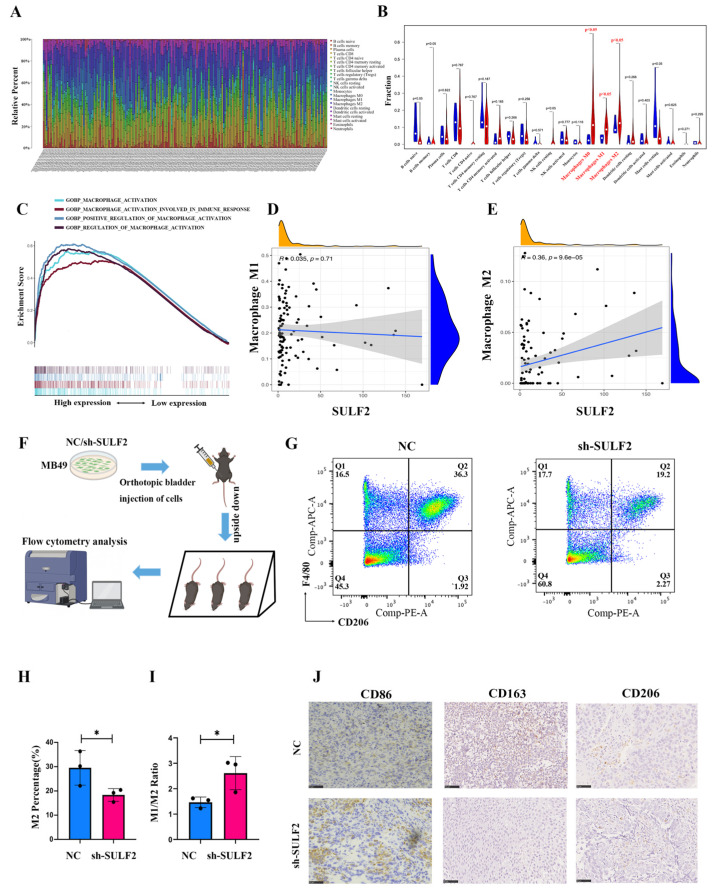
SULF2 is closely related to macrophages in the BCa microenvironment. (**A**,**B**) TCGA database analyzes the proportion of different types of immune cells in the microenvironment of BCa. (**C**) GSEA of SULF2 in STPH database. (**D**) SULF2 was negatively correlated with M1 macrophages. (**E**) SULF2 was significantly positively correlated with M2 macrophages. (**F**) Illustration of mouse orthotopic BCa model. (**G**–**I**) Flow cytometric analysis of mouse orthotopic BCa tissue expression of M2 macrophages (**G**), the M2 percentage (**H**), and M1/M2 ratio. (**I**,**J**) Representative images of CD163 and CD206 in mouse orthotopic BCa tissue by IHC staining. Scale bar = 100 μm. * *p* < 0.05.

**Figure 6 cancers-15-00131-f006:**
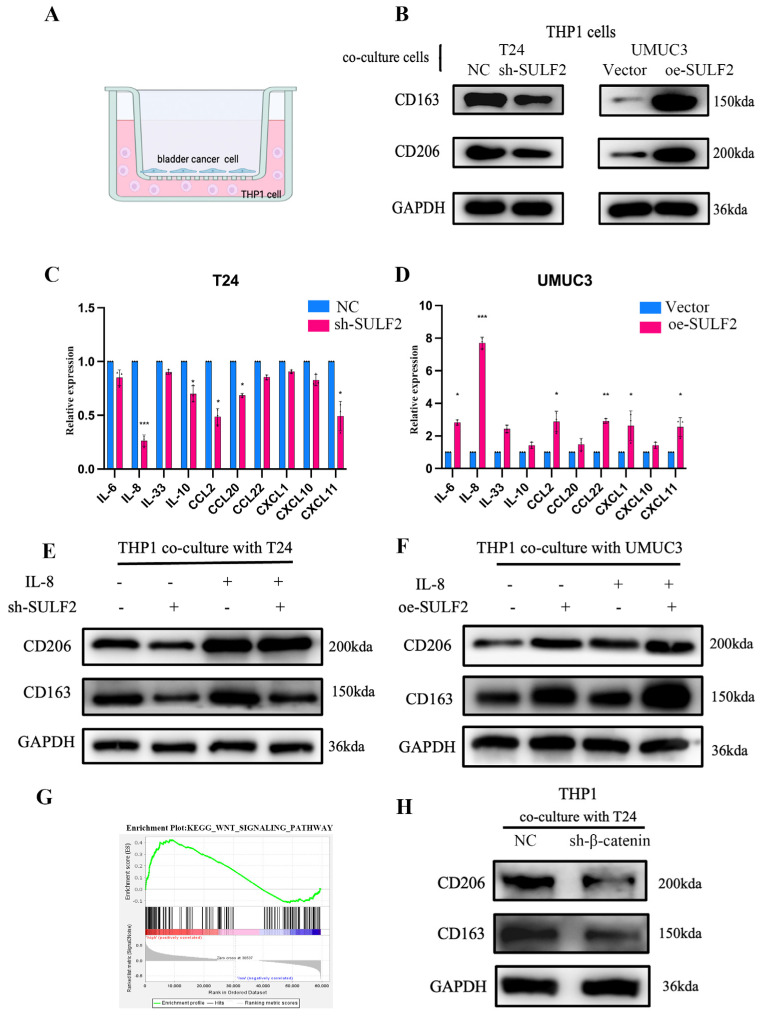
SULF2 promotes secretion of IL-8 through β-catenin. (**A**) An illustration of coculture of THP1 with BCa cells. (**B**) After coculture of THP1 with knockdown or overexpressing SULF2 cells, CD163 and CD206 expression in THP1 cells was detected by western blot. (**C**,**D**) Multifactor ELISA showed that knockdown or overexpression of SULF2 has most significant effect on IL-8 among other interleukins. (**E**,**F**) After exogenous addition of IL-8 to the coculture system, the expressions of CD163 and CD206 were detected by western blot. (**G**) GSEA analysis showed that SULF2 is significantly positively related to Wnt/β-catenin. (**H**) After knockdown of β-catenin in the coculture system, the expressions of CD163 and CD206 were detected by western blot. Data are represented as mean ± SD of three experiments. * *p* < 0.05, ** *p* < 0.01, *** *p* < 0.001.

**Figure 7 cancers-15-00131-f007:**
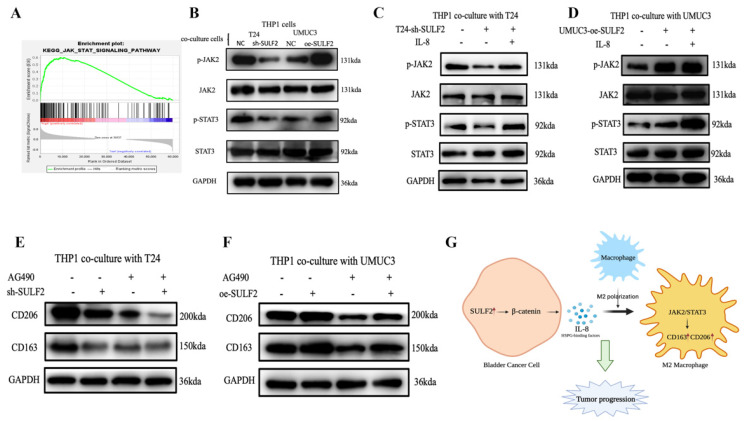
IL-8 promotes the polarization of macrophages through the JAK2/STAT3 pathway. (**A**) GSEA analysis showed that SULF2 was related to the STAT pathway in TCGA database. (**B**) The phosphorylation of JAK2 and STAT3 was detected by western blot after coculture of THP1 with BCa cells. (**C**,**D**) The phosphorylation of JAK2 and STAT3 was detected by western blot after exogenous addition of IL-8 to the coculture system. (**E**,**F**) The phosphorylation of JAK2 and STAT3 was detected by western blot after addition of the STAT3 inhibitor AG490 to the coculture system. (**G**) An illustration of the mechanism of SULF2 in the BCa microenvironment.

## Data Availability

The data that support the findings of this study are available from the corresponding authors upon reasonable request.

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
