# Peer review of "Sulfatase 2 Affects Polarization of M2 Macrophages through the IL-8/JAK2/STAT3 Pathway in Bladder Cancer"

_cancers, 2022, doi:10.3390/cancers15010131_

Round 1
Reviewer 1 Report
This is an interesting study. Zhang et al. explored the role of SULF2 in the microenvironment of bladder cancer and found that SULF2 highly expressed in bladder cancer can affect the polarization of macrophages. The manuscript is well organized. The experimental results of the study are adequate and can support the authors' conclusions. Some minor issues still need to be noticed:
1) Some figures are not marked with scale bar, such as Fig.2A, Fig.2G and Fig.S1.
2) In the Discussion section, The clinical application prospect of SULF2 should be discussed.
3) In the manuscript, the writing about “M2 macrophages” is inconsistent, such as M2 macrophages, macrophages M2, and M2-type macrophages.
4) The method section lacks the experimental description of ELISA method in this study, please describe in detail.
5) The histogram in the figures should maintain a unified style, such as Fig2C and 2F, Fig3A and 3H, etc.
6) In Line397 "SULF2 not only exerts its own biological functions" should be specified.
7) Research involving people should provide ethical approval and ethical number.
Author Response
This is an interesting study. Zhang et al. explored the role of SULF2 in the microenvironment of bladder cancer and found that SULF2 highly expressed in bladder cancer can affect the polarization of macrophages. The manuscript is well organized. The experimental results of the study are adequate and can support the authors' conclusions. Some minor issues still need to be noticed:
- Some figures are not marked with scale bar, such as Fig.2A, Fig.2G and Fig.S1.
Reply:Thank you for your comments. We noticed that we did not label the scale in Fig2A, 2G,Fig4A,4B and FigS1. We added the scale to the revised manuscript.
Changes in the figures:
- In the Discussion section, The clinical application prospect of SULF2 should be discussed.
Reply:Thank you for your suggestion. We believe it is necessary to add the potential application of SULF2 in the manuscript. Therefore, in the discussion and conclusion section of the revised manuscript, we have added a discussion on the potential application and clinical translation of SULF2.
Changes in the manuscript:
Line475-480:Our previous clinical study on SULF2 showed that IHC analysis and clinical prognostic follow-up of SULF2 expression in 203 patients with bladder cancer confirmed that SULF2 is highly expressed in bladder cancer and may be associated with lymphatic metastasis, which has clinical significance for evaluating prognosis. In addition, several studies have shown that adiponectin, an inhibitor of SUFL2, may be a potential tumor therapeutic drug. The role and mechanism of adiponectin in bladder cancer still need further verification.
Line533-534:In the future, the screening and validation of SULF2 inhibitors may require further investigation.
- In the manuscript, the writing about “M2 macrophages” is inconsistent, such as M2 macrophages, macrophages M2, and M2-type macrophages.
Reply:Thank you for your comments. We have revised the manuscript and uniformly written it as "M2 macrophages" in the article.
- The method section lacks the experimental description of ELISA method in this study, please describe in detail.
Reply:Thank you for your comments. Due to an error, we omitted the description of Elisa in the method section. In the revised manuscript, we have added Elisa in the method section.
Changes in the manuscript:
Line145-178:Reagents and Enzyme-Linked Immunosorbent Assay (ELISA) Assay
Human recombinant interleukin 8 (IL-8; PeproTech, Rocky Hill, NJ) was dissolved in trehalose at a concentration of 100μg/ml and stored at −20°C. At the time of use, the final concentration of IL-6 was adjusted to 100 ng/ml in the appropriate medium. AG490 (MedChemExpress, Monmouth Junction, NJ), a tyrosine kinase inhibitor that can inhibit the signal transducer and activator of transcription 3 (STAT3) signaling pathway, was dissolved in dimethyl sulfoxide at a concentration of 5 mg/ml, stored at −20°C, and protected from light. The THP-1 cells were seeded at a density of 3×106 cells/flask and exposed to 100 ng/ml phorbol 12-myristate 13-acetate (PMA; Sigma-Aldrich) for 48 hours to obtain macrophage-like differentiated THP-1 cells. Then the medium containing PMA was replaced with fresh medium to obtain resting state of macrophages (M0). Next, to differentiate into M1 phenotype we added 20 ng/ml IFNγ and 1 mg/ml LPS (Sigma-Aldrich), and for M2 phenotype we added 20 ng/ml interleukin 4(IL-4; PeproTech, Rocky Hill, NJ).
The Human Cytokine Elisa kit(Zorin Biological,Shanghai China) was used to detect the concentration of the supernatant cytokines in the co-culture system. Cytokine and chemokine levels were determined with standard curves developed for each experiment according to the standards provided by the manufacturer. The relative expression levels were calculated from the experimental results.
- The histogram in the figures should maintain a unified style, such as Fig2C and 2F, Fig3A and 3H, etc.
Reply:Thank you for your comments. We have adjusted the style of the statistical histogram to unify in the revised manuscript (Fig2C and 2F, Fig3A, 3H, fig6C and fig6D).
- In Line397 "SULF2 not only exerts its own biological functions" should be specified.
Reply:Thanks for your comment. We have changed this sentence in the revised manuscript
Changes in the manuscript:Line482-483:SULF2 not only changes the function of HSPG by regulating 6-O-sulfation, but also plays an important role in the occurrence and development of tumors, such as participating in tumor cell cycle, proliferation, apoptosis, and other biological functions.
7) Research involving people should provide ethical approval and ethical number.
Reply:Thanks for your comments, we have added the ethical number to the manuscript.
Changes in the manuscript:Line77-80:Ninety BCa tissues were collected from patients who underwent either TURBT or radical cystectomy at the Shanghai Tenth People’s Hospital (STPH) from November 2019 to September 2020. Prior informed consent was obtained from the patients(SHSY-IEC-4.1/19-120/01).

Reviewer 2 Report
In this article, Zhang et al. demonstrated that Sulfatase2 (SULF2) is overexpressed in bladder cancer patients and associated with poor overall survival. Mechanistically, authors have found that overexpression of SULF2 promotes the cell proliferation, migration, and invasion in vitro and slow down the tumor growth in- vivo. In addition, authors have demonstrated that SULF2 expression in tumor cells modulates the tumor microenvironment. They have found that SULF2 promotes the Macrophage polarization to the M2 -like phenotype through IL-8/AK2/STAT3 pathway.
Comments-
1- In figure 5 G, authors have shown that CD206 positive macrophages are less in Sh-SULF2 tumor mice compared to NC. CD206 is a commonly used marker for M2 -like macrophages. Can you add markers like Arginase-1 for M2 and NOS2, MHCII, and CD86 for M1 to validate the finding with additional evidence?
2- In Figure 5G- authors, is there any difference in overall macrophage infiltration in tumors?
3- Please provide the densitometry analysis western blot 6B, E, F and figure 7 or please re-confirm the finding with a other quantitative methods like flow cytometry.
5- Authors, did you see T and other immune cells in WT and Sh-SULF2 tumor microenvironment? If yes, could you provide data for it?
Reviewer 3 Report
This manuscript describes roles of Sulf-2 expression in bladder cancer and effects on macrophages. The manuscript is technically well performed and I have no criticism.
Author Response
Thanks for your comment.
Reviewer 4 Report
In this manuscript, the Zhang W and colleagues have demonstrated that SULF2 is significantly upregulated in bladder cancer. Using SULF2 knockdown and over-expression systems, they showed that SULF2 is important in promoting tumor cell proliferation, migration and invasion. Furthermore, these findings were also validated using both subcutaneous and metastasis mouse models. With co-culture experiments, they showed that SULF2 induces macrophage polarization by secreting IL-8, which in turn activates STAT3 pathway in macrophages. This study is important as it demonstrates the role of SULF2 in modulating immune cells in tumor microenvironment in context of bladder cancer. The comments for the manuscript are as listed
1. It is recommended that the authors also include some information about SULF1 as well in the background and introduction section.
2. In the cell lines and reagents section, the authors have described a cell line named 5637 but the data pertaining to that cell line is not available.
3. For the mouse models,
a. It will be very helpful to divide the segments into orthotopic model and syngeneic model for better clarification for readers.
b. The authors have utilized over-expression system as well for the orthotopic model so please include that in the description as well.
c. Also, were there any changes in body weight of mice in different studies over the duration of studies because of SULF2 knockdown or over-expression. Please provide data as supplemental figures
4. In result section, please expand MIBC and NMIBC (line 218)
5. For figure 1, the authors have described upregulation of SULF2 in tumor tissue but that is comprised of tumor cells and stroma. Can the authors specifically show that the upregulation was contributed by tumor cells given the fact that SULF2 is also a secreted protein and can be present in other stromal cells?
6. Please check the legend for figure 1, 3, and 4, which describes the number of repeats for the experiments and also significance. Were the animal experiments also repeated 3 times?
7. Lines 287-288, please describe findings for both over-expression and knockdown system for the metastasis animal model.
8. Figure 4H, quantification of the metastasis model depicts data points from 6 mice each for both the over-expression and knockdown systems but the methodology section describes use of 3 mice/ group for metastasis model. Please verify.
9. Line 309-310, figure 5G does not describe the syngeneic model but 5F does. Please update the text accordingly. Provide proper cross-references to the figures and results for this section.
10. Did the authors quantify by ELISA as to how much SULF2 is secreted by the tumor cells and also do these macrophages also produce SULF2?
11. Did the authors use STAT3 inhibitor (static) or Wnt pathway inhibitors to validate the findings about the pathways involved? It would also be helpful to see if IL8 was indeed upregulated in plasma or IHC for STAT3 and/ or beta-catenin in tumor tissue of mice used in syngeneic model.
